# Perceptions of and Practical Experience with the National Surveillance Centre in Managing Medicines Availability Amongst Users within Public Healthcare Facilities in South Africa: Findings and Implications

**DOI:** 10.3390/healthcare11131838

**Published:** 2023-06-24

**Authors:** Marco F. Falco, Johanna C. Meyer, Susan J. Putter, Richard S. Underwood, Hellen Nabayiga, Sylvia Opanga, Nenad Miljković, Ephodia Nyathi, Brian Godman

**Affiliations:** 1Department of Public Health Pharmacy and Management, School of Pharmacy, Sefako Makgatho Health Sciences University, Garankuwa, Pretoria 0208, South Africa; hannelie.meyer@smu.ac.za (J.C.M.); brian.godman@strath.ac.uk (B.G.); 2United States Agency for International Development Global Health Supply Chain—Technical Assistance, Hatfield, Pretoria 0083, South Africa; sue.putter@za-scta.com (S.J.P.); stirling.underwood@health.gov.za (R.S.U.); 3South African Vaccination and Immunisation Centre, Sefako Makgatho Health Sciences University, Garankuwa, Pretoria 0208, South Africa; 4Management Science Department, Strathclyde Business School, University of Strathclyde, 199 Cathedral Street, Glasgow G4 0QU, UK; h.nabayiga@strath.ac.uk; 5Department of Pharmacology, Clinical Pharmacy and Pharmacy Practice, University of Nairobi, Nairobi P.O. Box 30197-00100, Kenya; sopanga@uonbi.ac.ke; 6Institute of Orthopaedics Banjica, University of Belgrade, 11000 Belgrade, Serbia; nenad.hedren@gmail.com; 7Affordable Medicine Directorate, National Department of Health, Pretoria 0001, South Africa; ephodia.nyathi@health.gov.za; 8Department of Pharmacoepidemiology, Strathclyde Institute of Pharmacy and Biomedical Sciences, University of Strathclyde, Glasgow G4 0RE, UK

**Keywords:** public healthcare sector, South Africa, National Surveillance Centre, medicines availability, health systems strengthening, visibility and analytics network-operating model, medicine value chain, RxSolution, Stock Visibility System

## Abstract

The introduction of the National Surveillance Centre (NSC) has improved the efficiency and effectiveness of managing medicines availability within the public healthcare system in South Africa. However, at present, there is limited data regarding the perceptions among users of the NSC and challenges that need addressing. A descriptive quantitative study was performed among all registered active NSC users between August and November 2022. Overall, 114/169 users responded to a custom-developed, self-administered questionnaire (67.5% response rate). Most respondents used the Stock Visibility System (SVS) National Department of Health (NDoH) (66.7% for medicines and 51.8% for personal protective equipment (PPE) or SVS COVID-19 (64.9% for COVID-19 vaccines) or RxSolution (57.0% manual report or 42.1% application programming interface (API)) for reporting medicines, PPE, and COVID-19 vaccines to the NSC and were confident in the accuracy of the reported data. Most respondents focused on both medicines availability and reporting compliance when accessing the NSC, with the integrated medicines availability dashboard and the COVID-19 vaccine dashboard being the most popular. The respondents believed the NSC allowed ease of access to data and improved data quality to better monitor medicines availability and use. Identified areas for improvement included improving internet connectivity, retraining some users, standardising the dashboards, adding more data points and reports, and expanding user adoption by increasing licence limits. Overall, this study found that the NSC in South Africa provides an effective solution for monitoring and improving medicines availability.

## 1. Introduction

A key part of the overall performance of a health system is the medicine value chain, with the availability of medicines playing a key part in improving the care of patients with both infectious and non-infectious diseases [1,2,3]. However, various investigations have raised concerns about the availability of medicines in the public healthcare system in South Africa, which needs to be addressed to improve the care of patients in the country [4,5,6,7]. This mirrors a similar situation across multiple countries and continents [8,9,10].

In September 2015, in response to findings from the Auditor General regarding the management of pharmaceuticals during the years 2011/12 to 2014/15, the South African Minister of Health appointed an advisory team for the security of pharmaceuticals and related commodities. The team comprised of local and international experts, including members from the World Health Organization (WHO), Health Action International (HAI), and Supply Chain Management Services (SCMS). The team was tasked to understand the factors contributing to medicines availability challenges. They subsequently identified that the specific challenges affecting medicines availability and access across the country highlighted the need for a comprehensive, evidence-based response to address ongoing concerns [4,11,12]. In South Africa in 2017, the Stop Stock-outs 4th National Survey Report was published. This report stated that the supply of medicines was currently provided by a fragile system and that stock-outs continued to be a challenge for healthcare facilities, necessitating urgent attention [13].

The challenges of providing an adequate supply of medicines and avoiding shortages are typically aggravated by the absence of measurable objectives. Alongside this, the lack of consolidated and accessible data regarding the current availability of medicines in the system, as well as a lack of medicine value chain performance measures [12,14]. In addition, different understandings of protocols among key stakeholder groups as well as poorly resourced health facilities exist. This includes insufficient human resources at the facility, district, provincial, and national levels, as well as concerns with the level of training of key staff members. As a result, developers and implementers of policies can be constrained, acting with little or no evidence and inadequate resources to design interventions to address known problems. This is also a concern for patients where issues and challenges with procurement strategies can exacerbate medicine shortages, potentially impacting their care [15].

The goal of the Visibility and Analytics Network (VAN) operating model is to provide end-to-end visibility of the supply chain. It is a set of supply chain capabilities with a shared service at a central point, built on people, policies, processes, and technology [16,17,18,19,20]. In South Africa, several innovations have recently been developed and implemented to improve medicines availability and access in the public system. In 2016, the National Department of Health (NDoH) and the United States Agency for International Development (USAID) via the Global Health Supply Chain Program—Technical Assistance (GHSC-TA) collaboratively developed the National Surveillance Centre (NSC), a new early warning system regarding medicines availability that was initially donor funded [21]. This is a web-based performance-monitoring and evaluation tool, used to provide visibility of medicine stock levels, at present, across all levels of the medicine value chain. Alongside this, it seeks to improve the availability of medicines across all provinces in South Africa. The use of nationally agreed key performance indicators allows medicines availability data from hospitals and clinics, pharmaceutical depots, and suppliers of medicines to be visualised on the NSC, consequently, providing a holistic view of the availability of medicines and other related commodities throughout the South African public healthcare medicines supply chain. As a result, they allow the user to predict where shortages are likely to occur and act proactively rather than reactively [6].

This is important in the management of infectious diseases. There are concerns over the supply of antibiotics across facilities in South Africa and Africa. This should be addressed along with enhancing the adherence to recommended antibiotics, especially those with less resistance potential, such as those in the WHO ‘Access’ group to reduce high levels of antibiotic resistance in South Africa, which increases morbidity, mortality, and costs [22,23,24,25,26,27]. Good supplies of COVID-19 vaccines were also needed at the time of the survey to control the spread of SARS-CoV-2, as well as decrease symptoms of those with COVID-19 [28,29,30]. During the course of the pandemic, there were concerns over vaccine hesitancy in South Africa and the NDoH did not want to complicate this by compromising the supply of the vaccine [31,32]. The provision of public health measures, including personal protective equipment (PPE), is also important to slow the spread of the virus and the subsequent implications on future morbidity and mortality rates [33].

The rise in deaths in South Africa in recent years from cardiovascular disease has also resulted in a number of initiatives undertaken by government and other organisations being developed [34,35,36]. Ensuring the adequate availability of recommended medicines, alongside tackling problems with adherence to prescribed medicines, should also be a priority, especially in the public healthcare system, to reduce rising mortality rates [6,37].

The NSC was deployed in 2016, and as of 14 October 2021, 3826 facilities have reported their medicine stock levels via the Stock Visibility System (SVS) as well as other electronic stock management systems, including RxSolution [13]. With the implementation of the NSC throughout South Africa, information is provided to users in each province about medicine stock levels at each facility in their province. The result is reduced medicine wastage and the improved availability of medicines compared with the historic situation [38]. This system builds on initiatives in hospitals across Africa to monitor the availability and use of medicines to improve the care of patients, especially in priority disease areas [39,40,41]. However, it is still unknown what influence the NSC system has had on how users of the NSC intervene when there are issues regarding the availability of key medicines, and whether the NSC has provided them with the necessary data to identify and resolve medicine availability issues. This is important, as we have seen that re-designing systems can improve access to medicines, as well as reduce stock-outs and costs in public healthcare systems among African countries [42,43,44,45].

Consequently, this study aims to address this gap and determine NSC users’ perceptions of and experience with the implementation and use of the system in practice, to manage and improve medicines availability within public healthcare facilities across South Africa. Furthermore, we aim to determine the experience of users with training received on the NSC and to identify any future training needs. To the best of our knowledge, this is the first study to describe the perceptions of active users of the NSC as well as provide additional insights from a management perspective following a recent study assessing the knowledge, attitudes, and practices of healthcare professionals (HCPs) at the primary healthcare (PHC) facility level on the use of the SVS for reporting the availability of medicines in the country [12]. The findings from the present study can be used to provide additional guidance to the NDoH in South Africa to further improve the system, if pertinent.

## 2. Materials and Methods

### 2.1. Study Setting and Design

The NSC, a web-based performance-monitoring and evaluation tool, is used to provide visibility of medicine stock levels across all levels of the medicine value chain in the public healthcare sector in South Africa [21]. The public healthcare system is important in South Africa and provides care to the vast majority of patients [6]. However, it is underfunded, at present [46].

The design of the study was descriptive and quantitative, using an electronic survey amongst the users of the NSC. The target study population included all registered NSC users having accessed the NSC in the six months prior to the survey.

The NSC users were categorized into three user levels, as shown in Table 1. National users were based at a national level, monitoring medicines availability at a country level. They mainly comprised national management, programme managers, and contract managers. Provincial users were typically based at a provincial level and monitored medicines availability in their respective provinces. They generally comprised district/subdistrict pharmacy managers. Their role was facilitating and monitoring patient access to medicines at facilities. The support partners, with approval by the NDoH, were given access to the NSC in South Africa to assist with improving medicines availability throughout the country.

The NSC users were further categorised into subgroups called ‘user groups’, which indicated the department, province, or non-governmental organisation (NGO) where each of the NSC users were employed. At the time of the study, there was a total number of 268 registered NSC users, of whom 169 were active users having accessed the NSC within the 6 months prior to the survey.

An all-inclusive sample was used. All registered users on the NSC user database, who were active within the last 6 months prior to data collection, were invited to participate in the survey. Registered users who had not accessed the NSC within the previous 6 months, and any user who did not agree to participate in the survey, were excluded from the study.

### 2.2. Data Collection

#### 2.2.1. Data Collection Instrument

The data collection instrument for this survey was a self-administered questionnaire completed online in English, as the NSC is only available in English and this is the official medium of communication in the workplace within the NDoH in South Africa (Appendix A).

The questionnaire was developed by the principal author (MFF), specifically for the purpose of this study. This was based on considerations of how the NSC could impact on users’ day-to-day work in monitoring medicines availability and facilitate patients’ access to medicines. Alongside this, it was acknowledged that the NSC has multiple data sources that it uses to generate the multiple dashboards in addition to issues of training and overall feedback. This approach was used before when generating context-specific questionnaires [32,47].

To enhance the robustness of the developed questionnaire and for content validity, the draft questionnaire was reviewed by two experts in the field for its content and language. Minor changes were subsequently made to the questionnaire to improve the understanding of the questions and to reduce any ambiguity. Following this, the questionnaire was pretested among two people, including one of the co-authors.

The final questionnaire (Appendix A) included close-ended, rating scale, and open-ended questions to collect the data on the following variables: sociodemographic information, systems used to report to the NSC, how users access and use the system, training, and the NSC system itself alongside recommendations for improvement.

The questionnaire was subsequently made available using Google Forms, a Google web-based application, which is part of the Google suite, allowing the creation of surveys (https://docs.google.com/forms/u/0/, accessed on 11 June 2022). This application allowed the creation of a survey that could be completed online with multiple choices, short answers, and checkboxes, allowing for ease of completion and analysis. Depending on the answers, the questionnaire format allowed the respondents to be redirected to different sections of the form, streamlining the response process for respondents.

#### 2.2.2. Enrollment and Data Collection

A request was sent to the NSC development team for the list of all active NSC users and their email addresses. An email was subsequently sent to all active NSC users inviting them to participate in the survey. The email included information explaining the aim and objectives of the study, and a consent statement. The statement stipulated that, by completing the survey, respondents automatically provided consent to participate in the study.

Those who agreed to participate in the survey were able to click on an active link, which directed them to the online questionnaire. No personal identifiers were linked to responses, with each online completed response being assigned a unique timestamp. However, at the end of the questionnaire, there was an option for the respondents to share their contact details, if willing, should it be necessary for follow-up questions if clarity was needed.

The data collection occurred over a period of 10 weeks between 24 August and 2 November 2022. To ensure a good response rate, reminders were sent every two weeks to all active NSC users to remind them to complete the survey, if they had not already done so (Figure 1).

### 2.3. Data Management and Analysis

Once all the data were electronically collected using Google Forms, the responses were downloaded into Microsoft Excel^®^ for Microsoft 365 MSO (Version 2212 Build 16.0.15928.20196) 64-bit for analysis. To ensure the validity of the data, they were firstly checked for accuracy and cleaned. This process involved ensuring that all fields were filled out correctly, highlighting outliers or errors.

Due to the descriptive design of this study, Microsoft Excel^®^ was used to summarise the dataset by calculating frequencies and percentages for all the variables. Free text responses to open-ended questions, which were also exported to Microsoft Excel^®^, were firstly read by MFF to achieve an understanding of the data. This was followed by identifying and assigning codes to categories evident from the responses. These categories and codes were subsequently discussed with JCM, agreed upon, and then used by MFF to code all open-ended responses in Microsoft Excel^®^. Final coding was verified by JCM, including a discussion of any discrepancies with MFF and reaching an agreement on the coding. This process resulted in a set of categorical data for the free text responses, which allowed the calculation of frequencies and percentages for each category (Figure 1).

## 3. Results

### 3.1. Response Rate, Sample Population, and Demographics

All 169 presently active NSC users were invited to participate in the study, of whom 114 responded, resulting in a response rate of 67.5%. The detailed response rate is shown in Appendix B, Table A1, with the detailed demographic characteristics of the respondents summarized in Appendix B, Table A2. Most respondents were females (64.9%) with more females represented in the older age groups (39.2% aged 40–50 years). Males, in contrast, were more represented in the younger age groups with 40.0% of males aged 30–40 years old. Overall, females were more prevalent in the study, particularly among White respondents and in the 40–50-year-old age group. Males were more represented among Black respondents (52.5%), particularly in the 30–40-year-old age group.

The respondents held a range of qualifications, with the majority having a Bachelor’s degree or Master’s qualification. The respondents also held a range of positions. This included managerial roles in pharmaceutical services, district and sub-district pharmacists, hospital pharmacists, policy specialists, information technology (IT) support, programme managers, and others. In terms of their number of years in their current position, the largest group (34.2%) had been in their position for 3–5 years, followed by those in their position for 10 years or more (31.5%).

Most respondents stated that their main roles within the NSC related to reporting on medicines availability, reporting compliance levels, distribution of NSC information, and management of stock-outs. Other roles included redistributing stock, forecasting and planning, TEE (tenofovir, emtricitabine, efavirenz) to TLD (tenofovir, lamivudine, dolutegravir) transition, procurement decisions, and compiling reports (Table 2).

### 3.2. Stock Management Systems

There were two different instances of SVS allowing for different devices to be used for medicines and PPE, and another for COVID-19 vaccines. Most respondents used SVS NDoH or SVS COVID-19 for reporting to the NSC. The respondents also used other systems, including RxSolution, which has two methods of reporting, manually using a report generated by RxSolution and emailed to the NSC and an automated reporting system, RxSolution application programming interface (API). Warehouse management systems (WMSs), including the Medical Supplies Administration System (MEDSAS), Pharmaceutical Distribution System version 10 (PDSX), Oracle, and Government Commerce (gCommerce), are systems used at pharmaceutical depots (Figure 2). Figure 3 provides details regarding how the respondents perceived the accuracy of the data within the systems, at present.

### 3.3. Assessing Current Usage of the National Surveillance Centre

Most respondents accessed the NSC data weekly (87/165; 52.7%), followed by daily (31/165; 18.8%), monthly (25/165; 15.2%), or occasionally (22/165; 13.3%). A laptop (94/165; 57.0%) was used by most respondents, followed by the use of a desktop (38/165; 23.0%), mobile telephone (24/165; 14.6%), or tablet (9/165; 5.5%) (Appendix B, Table A3), with a minority (36.8%) using multiple devices.

The key areas that respondents focused on when accessing the NSC included both the availability of medicines and reporting compliance levels (59.7%), medicines availability and medicine stock levels (32.5%), reporting compliance alone (4.4%), accessing all areas (2.6%), or the minimum/maximum dashboard (0.9%).

Details regarding the dashboards that the respondents accessed most frequently in the NSC are presented in Table 3. The integrated medicines availability dashboard, which visualises all data sources into one view, was the most frequently used among 83.3% of the respondents. This was followed by the COVID-19 vaccine dashboard and the integrated medicines availability dashboard for trend analysis.

### 3.4. National Surveillance Centre Training

Figure 4 details the responses from the respondents regarding current NSC training they have experienced.

Regarding the adequacy of training provided at present, 27.4% of the respondents strongly agreed that this was adequate, 51.6% believed that training was adequate, and 10.5% were neutral, while 3.2% believed that training was not adequate and 7.4% strongly disagreed that training was adequate.

Of those who had not yet received training, the reasons given were that training had not been provided (26.3%), they could not attend the training on the dates provided (26.3%), they did not need any training because they could independently navigate and understand the system (42.1%), or they were recently appointed (5.3%). Table 4 contains details of the training requirements that respondents answering the question regarding what training they would like to receive to further improve their ability to manage medicines availability within the public healthcare system.

### 3.5. Confidence and Accuracy of the Various Dashboards and Recommendations

Building on Figure 3, the integrated medicines availability dashboard was rated the highest in terms of satisfaction and accuracy (52.4% highly satisfied; 27.2% very highly satisfied; 53.4% highly accurate; 14.6% very highly accurate) with similar findings for the COVID-19 dashboard (45.3% highly satisfied; 28.0% very highly satisfied; 44.0% highly accurate; 21.3% very highly accurate).

The COVID-19 PPE dashboard was rated the lowest in both categories compared to the other two dashboards, with only 36.8% of the respondents stating they were highly satisfied and 42.1% believed the content was highly accurate (Appendix B, Table A4). This dashboard though was also the least accessed and monitored. This could have been because PPE during the COVID-19 pandemic was seen in many provinces as out of the scope of pharmaceutical services. Table 5 contains details of the functionality of the various dashboards favoured by the respondents, with Table 6 providing recommendations for the future.

When comparing the three dashboards (integrated medicine, COVID-19 vaccine, and COVID-19 PPE), all three were widely used and observed to provide valuable information and tools. However, areas for improvement were identified.

Suggested improvements for the COVID-19 vaccine dashboard included adding data on the number of individuals vaccinated, capturing and displaying reasons for stock lost, allowing users to search for information by facility name or code, batch management, and enhancing data capturing accuracy.

The integrated medicines availability dashboard was mainly used for monitoring medicine stock levels for redistribution, monitoring reporting compliance, and analysing trends over time. Users mostly found this dashboard easy to access data, with useful visual explanations for a rapid overview of medicines availability and the ability to share the data with colleagues, which helped build better relationships and communication. Suggested areas for improvement included improving the accuracy and timeliness of the data, integrating the datasets to provide a single view, addressing data anomalies between the different views, improving filters, including Average Monthly Consumption (AMC) data, improving reporting accuracy and compliance by facilities and adding an expiry date report.

### 3.6. Consolidated Feedback Regarding the NSC

Most of respondents rated the overall usefulness of the NSC and the overall experience as high or very high, with the purpose of the NSC being a central point of visibility with little manipulation of the data to obtain the information required. The respondents found the NSC easy to navigate, with only a few stating some difficulty. Overall, the introduction of the NSC significantly benefited the users by improving their ability to proactively identify stock challenges and redistribute stock accordingly in their area, use the data for reporting purposes, and monitoring the performance of suppliers.

The NSC for many of the respondents streamlined or completely replaced a previous tedious manual process of monitoring medicines availability. Three quarters (75%) of the respondents monitored medicines availability before the implementation of the NSC. However, this was mostly undertaken manually using facility stock management systems, facility site visits, or waiting for facilities to actually report stock issues, including overstocking or stock-outs. The use of the NSC saved time by replacing these processes and by providing improved access to accurate data. This allowed the respondents to improve the timeliness and quality of interventions. The remaining 25% of the respondents indicated that they did not monitor medicines availability before the NSC as it was too difficult and time-consuming. This activity was a manual task with the associated time implications; alternatively, their role was different and did not require monitoring of medicines availability.

The most-extracted data by the respondents from the NSC was medicine stock level data allowing for interventions as well as assessing medicines availability and reporting compliance. The data was mainly used for reporting purposes and sharing this with others at the facility level, with 82% of the respondents also indicating this data was used for creating presentations. The main audience for reports or presentations were managers, including program managers, district, provincial, and national management. However, some data was extracted outside of the NSC to add other data sources to the NSC data. This allowed more flexibility in manipulating the data to the specific needs of the respondents or the format they required.

Overall, 66% of the respondents stated that the NSC had a positive impact on medicines availability. This was due to the fact the data were easy to access, allowing proactive interventions to redistribute excess stock to where it was needed. Table 7 highlights the ways the respondents indicated that the NSC improved medicines availability and their monitoring. However, 7% of the respondents stated the impact on medicines availability data from the NSC was limited, with 11% unsure if it had made an impact in reality.

The respondents also identified that the NSC had implications for stock management at the facility level as well. The respondents believed that the stock management processes improved (40%) due to the improvement of stock vigilance (16%). In addition, accountability (8%) at the facility level with supervisors having increased visibility of more accurate data was observed. These developments resulted in an improvement of stock rotation (16%) and medicine availability interventions (4%). However, a few respondents mentioned that the NSC had little actual impact to date (11%), due to supply challenges higher up the chain, or did not make a difference in their stock management.

Two thirds (65%) of the respondents indicated the NSC also affected areas outside of medicines availability. Areas of demand planning improved (11%) alongside supplier management (11%), assisting with transition periods or a pandemic (7%), improving relationships and communication beyond the area of support (4%). In addition, IT skills were forced to improve (4%) and there was improved work satisfaction and staff morale, with the highest commented areas being the importance of accurate reporting and compliance (20%).

Half (53%) of the respondents also stated that an escalation protocol was put in place in many facilities, districts, and provinces to monitor medicines availability and to rectify any issues identified. This protocol may involve communication with suppliers, redistribution of stock, and follow-up on outstanding orders. Some respondents also mentioned that the protocol was partially effective or could be improved, while others were not aware of its existence or had not yet observed its impact. Some respondents also noted that transport issues and a lack of written documents could be a challenge for executing the escalation protocol. Other challenges included internet connectivity, the quality of the data and frequency of data updates, and supporting systems at present (Table 8).

Recommendations to improve the NSC from the respondents were varied. In addition to recommendations for the various dashboards (Table 6) enhancing the accuracy of and confidence in any data provided, including its quality (Figure 3) as well as training needs (Table 4), the suggested improvements also included increasing the licence limit to allow for the greater adoption and use of the NSC. Adding more data points, including the ability to track batch and expiry dates, as well as improving medicine expiry date monitoring and the ability to monitor all products, will also help in the future. Alongside this, adding data on the AMC per item per facility, the standardisation of all the dashboards with their filters, and reviewing the categories of medicine will also help in the future to improve the monitoring of medicines and their availability.

## 4. Discussion

We believe this is the first study in South Africa to investigate user attitudes to the NSC, the ways in which this has improved medicines management, including stock availability and shortages, as well as provide guidance for future improvements. This study built on recent studies assessing the knowledge, attitudes, and practices of HCPs at the facility level on the use of the SVS for reporting the availability of medicines in South Africa, as well as concerns over stock management systems in PHCs [12,48]. In addition, we addressed concerns over supply chains across countries and potential ways to address them [1,3,49].

The overall response rate of 67.5% was encouraging, considering the small target population; however, we were aware that the response rate varied across the different user groups. This highlights the continual need for all user groups to have equal access to the NSC, as well as similar training in its use.

Moreover, the respondents appeared to have confidence in the accuracy of the data within these systems and were mostly satisfied in using the various dashboards. This was certainly enhanced by their overall satisfaction with the training they had received; however, there was room for improvement. The dashboards for which training was most requested were the minimum/maximum dashboard (69.1%), the demand planning dashboard (60%), the integrated medicines availability dashboard (60%), and the supplier management dashboard (50.9%).

It was clear from the findings of the research that the ways in which the NSC improved medicines availability and monitoring were due to the ease of access to the data, as well as improving data quality and quantity. Transparency in supply chains, as well as accuracy with respect to information on present stock levels and potential shortages, is crucial across countries to improve medicines availability [3,49]. However, the respondents in this study would like to see user adoption expanded and licences granted to more people to improve medicines availability and monitoring in the future. Furthermore, whilst many stakeholders receive reports that are generated and shared, they do not have access, at present, to the NSC, which can be a concern.

The functionality that the respondents favoured most, with respect to the NSC, was the ease of accessing the data. This allowed for informed decision making and the ability to improve stock levels and redistribution, which is vital going forward, especially for critical medicines, including antibiotics [23,24,25,49]. In the future, the respondents would like to see that each dashboard design standardised and additional reports as well as additional data points are added. User functionality could also be improved, for example, saving user preferences to speed up the filtering process or adding more patient data to be able to interpret the data more effectively, building on examples in other countries [50].

The challenges going forward in South Africa and beyond include improving internet connectivity and IT systems, as well as improvements in data quality and availability [1,50]. It was also clear that the respondents would like to see user adoption expanded, with licences given to more people to access the dashboards. These potential developments and their impact will be monitored in future research projects.

Overall, it is evident that from the results that people and processes are essential to a functional VAN operating model. Without fully trained staff members interpreting and applying the processes, the data have a reduced impact and cannot be maximally used. This is an appreciable concern in South Africa, given the rising burden of both infectious and non-infectious diseases, and the need to optimize patient treatment within the public healthcare system to reduce future morbidity and mortality [6,34,35,36,51,52].

We recognize there are a number of limitations with this study. Firstly, it is worth noting that our study was limited to active NSC users only, specifically those who had accessed the NSC within six months of the survey. We recognise that this resulted in a relatively small study population and subsequent small sample sizes limiting the generalizability of the results. However, to achieve the study objectives, our focus is on specifically targeting active NSC users. Secondly, we are aware of the potential of response and recall bias, which could not be completely avoided due to the nature of our self-reported questionnaire and the fact that the response rate varied across the different NSC user groups. Despite these limitations, we believe that the results of this study can provide guidance in terms of strengthening future systems to manage and improve medicines availability in the public healthcare system in South Africa in the future.

## 5. Conclusions

The importance of using technologies to collect, visualise, and interpret data to inform decision making is a critical component in the VAN operating model. The NSC in South Africa supports this function within the public healthcare sector by enhancing the ease of access to data on medicine stock levels at each healthcare facility. As a result, it has improved medicines management compared to historic paper-based and other systems.

However, there are still challenges that need to be addressed to further improve medicines management within the public healthcare system in South Africa. The identified areas include increased training and internet connectivity, as well as increasing the licence limit to increase NSC adoption in the public healthcare sector. Moreover, improving data quality and adding batch management data to enhance expiry date monitoring is essential.

## Figures and Tables

**Figure 1 healthcare-11-01838-f001:**
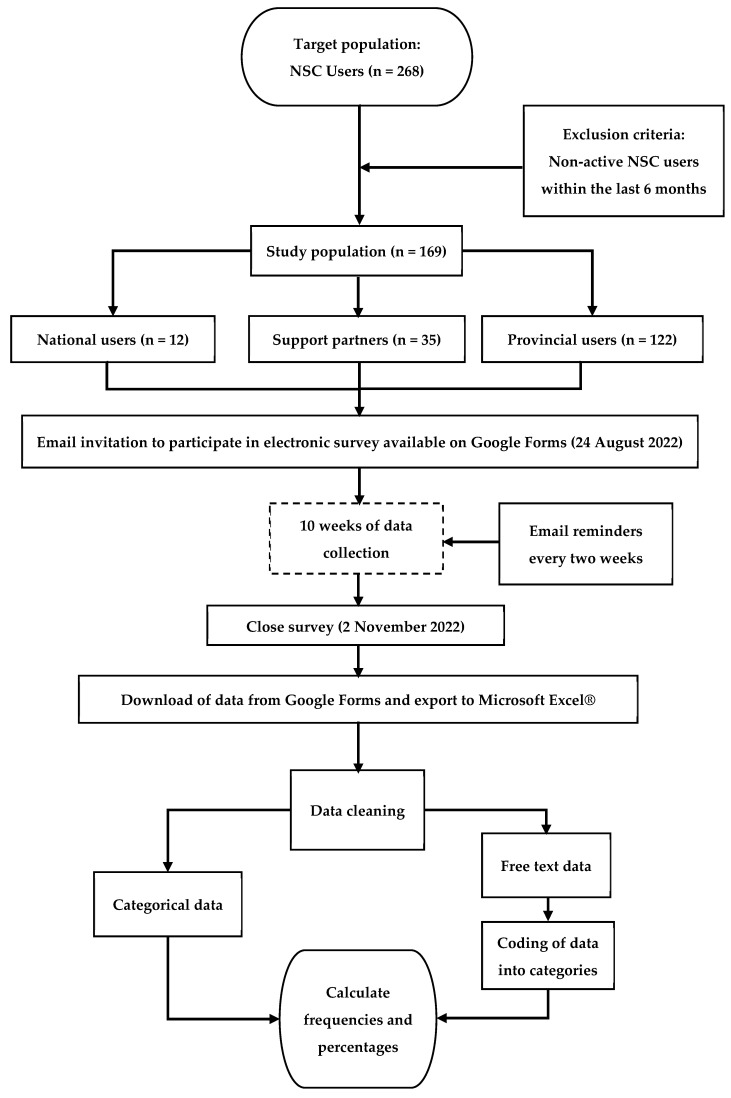
Methodology flow diagram.

**Figure 2 healthcare-11-01838-f002:**
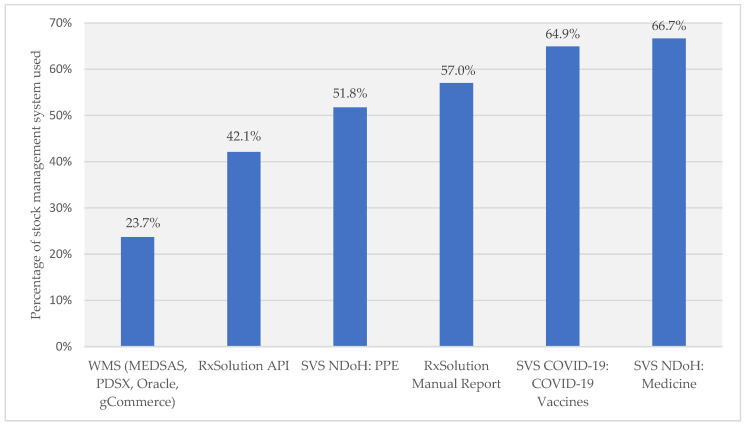
Electronic stock management systems being used at present, to report to the NSC. Nota Bene: WMSs: warehouse management systems; API: application programming interface; SVS: Stock Visibility System; NDoH: National Department of Health; PPE: Personal Protective Equipment; PDSX: Pharmaceutical Distribution System version 10; MEDSAS: Medical Supplies Administration System; gCommerce: Government Commerce.

**Figure 3 healthcare-11-01838-f003:**
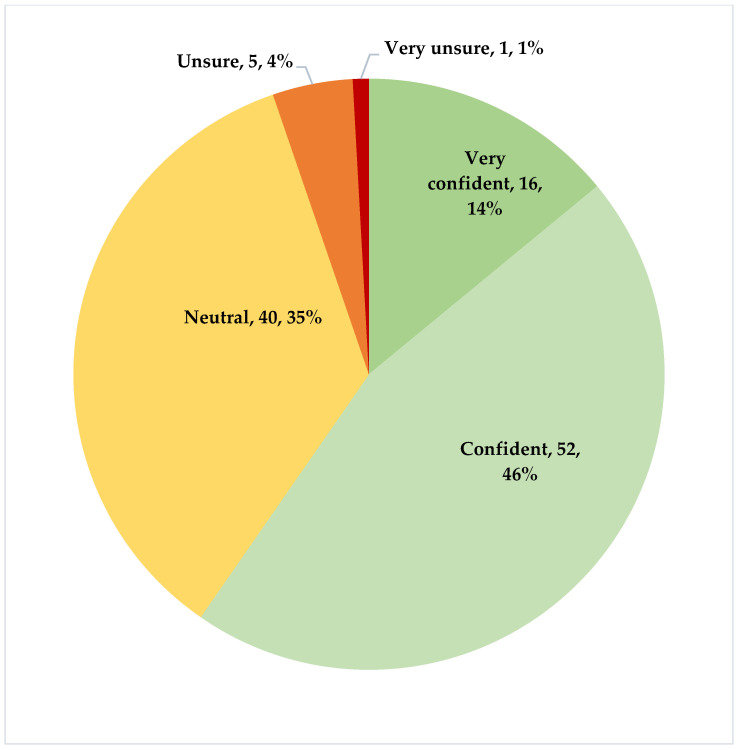
Perceived accuracy of the present systems (n = 114).

**Figure 4 healthcare-11-01838-f004:**
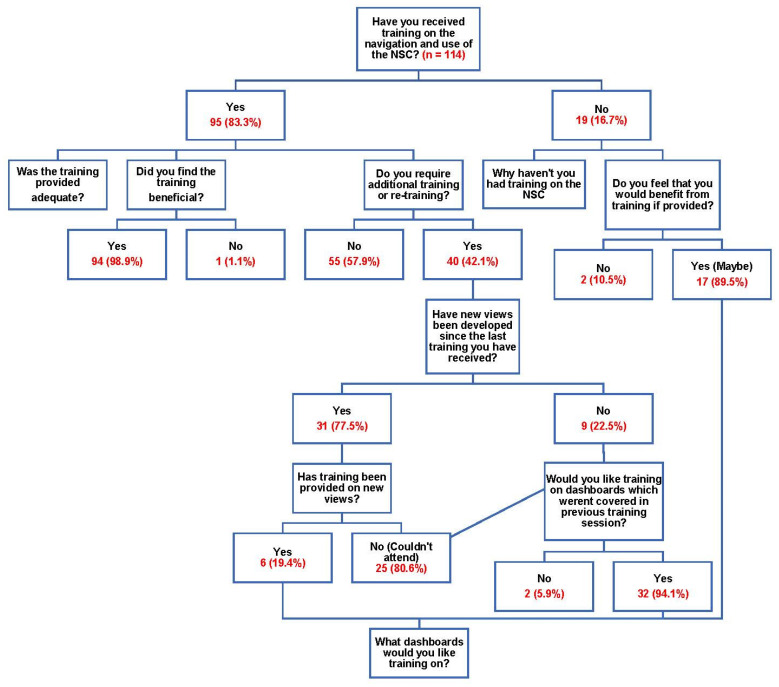
Responses regarding National Surveillance Centre training experienced, at present.

**Table 1 healthcare-11-01838-t001:** National Surveillance Centre active users (as of 17 August 2022).

User Level	User Group (Department, Province, or Support Partner)	Number of Active Users
National user	Administrator	1
National user	Affordable Medicines Directorate Management	1
National user	Contract Management Unit	7
National user	Personal Protective Equipment	1
National user	Programmes	2
Support partner	Africa Resource Centre	2
Support partner	Clinton Health Access Initiative	3
Support partner	Global Health Supply Chain—Technical Assistance	27
Support partner	Maternal, Adolescent and Child Health Institute	1
Support partner	United Nations International Children’s Emergency Fund	2
Provincial user	Eastern Cape	20
Provincial user	Free State	14
Provincial user	Gauteng	17
Provincial user	KwaZulu-Natal	28
Provincial user	Limpopo	7
Provincial user	Mpumalanga	10
Provincial user	North West	10
Provincial user	Northern Cape	12
Provincial user	Western Cape	4
**Total**		**169**

**Table 2 healthcare-11-01838-t002:** NSC users’ positions and roles in monitoring medicines availability.

Position	Stock-Out Management (n = 86)	Reporting Medicines Availability Status to Seniors (n = 83)	Distribute Information Obtained from NSC (n = 71)	Reporting Compliance Status to Seniors (n = 63)	Other (n = 14)	Total (n = 317)
District/sub-district pharmacist	37 (43.0%)	35 (42.2%)	32 (45.1%)	34 (54.0%)	0	138 (43.5%)
Support partner	12 (14.0%)	10 (12.1%)	12 (17.0%)	8 (12.7%)	11 (78.6%)	53 (16.7%)
Provincial pharmacist	11 (12.8%)	11 (13.3%)	13 (18.3%)	9 (14.3%)	1 (7.1%)	45 (14.2%)
Hospital pharmacist	12 (14.0%)	12 (14.5%)	6 (8.5%)	7 (11.1%)	0	37 (11.7%)
Depot pharmacist	7 (8.1%)	5 (6.0%)	1 (1.4%)	1 (1.6%)	0	14 (4.4%)
Head of pharmaceutical services	3 (3.5%)	3 (3.6%)	3 (4.2%)	2 (3.2%)	0	11 (3.5%)
Information technology support	1 (1.2%)	3 (3.6%)	3 (4.2%)	2 (3.2%)	0	9 (2.8%)
National pharmaceutical policy specialist	1 (1.2%)	1 (1.2%)	1 (1.4%)	0	0	3 (1.0%)
Contract manager	1 (1.1%)	1 (1.2%)	0	0	0	2 (0.6%)
Programme manager	1 (1.2%)	1 (1.2%)	0	0	0	2 (0.6%)
Administrator	0	0	0	0	1 (7.1%)	1 (0.3%)
Chief director	0	1 (1.2%)	0	0	0	1 (0.3%)
Director of affordable medicines directorate	0	0	0	0	1 (7.1%)	1 (0.3%)

**Table 3 healthcare-11-01838-t003:** Details of dashboards accessed by the respondents.

Dashboards Accessed	Responses (n = 114)
Integrated medicines availability dashboard	95 (83.3%)
COVID-19 vaccine dashboard	62 (54.4%)
Integrated medicines availability dashboard—trend analysis	62 (54.4%)
Historic dashboards—hospital level data only	32 (28.1%)
Historic dashboards—primary healthcare level data only	32 (28.1%)
COVID-19 PPE dashboard	31 (27.2%)
TEE/TLD transition dashboard	27 (23.7%)
Supplier management dashboard	27 (23.7%)
COVID-19 supply chain	25 (21.9%)
Rx API—facilities connectivity report	20 (17.5%)
Demand planning dashboard	17 (14.9%)
Minimum/maximum dashboard	2 (1.8%)
Usage of NSC by users	1 (0.9%)

Nota Bene: There were 114 completed responses regarding the details of dashboards accessed via suitable technologies; PPE: Personal Protective Equipment; TEE/TLD: Tenofovir, Emtricitabine, Efavirenz/Tenofovir, Lamivudine, Dolutegravir; Rx API: RxSolution Application Programming Interface; NSC: National Surveillance Centre.

**Table 4 healthcare-11-01838-t004:** Dashboards for which the respondents requested training.

Dashboards for the Required Training	Number of Respondents (n = 55)
Minimum/maximum dashboard	38 (69.1%)
Demand planning dashboard	33 (60.0%)
Integrated medicines availability dashboard	33 (60.0%)
Supplier management dashboard	28 (50.9%)
Integrated medicines availability dashboard—trend analysis	26 (47.3%)
Historic dashboards—primary healthcare level data only	22 (40.0%)
TEE/TLD transition dashboard	20 (36.4%)
COVID-19: vaccine dashboard	16 (29.1%)
Historic dashboards—hospital level data only	14 (25.5%)
COVID-19—supply chain	12 (21.8%)
COVID-19—PPE dashboard	6 (10.9%)
Pre-exposure prophylaxis	2 (3.6%)

Nota Bene: TEE/TLD: Tenofovir, Emtricitabine, Efavirenz/Tenofovir, Lamivudine, Dolutegravir; PPE: Personal Protective Equipment.

**Table 5 healthcare-11-01838-t005:** Functionalities favored by the respondents for each designated dashboard.

Dashboard Functionalities Favoured by Respondents	Integrated Medicine (n = 126)	COVID-19 (n = 79)	PPE (n = 42)	Total (n = 247)
Improved stock level and redistribution	42 (33.3%)	19 (24.1%)	16 (38.1%)	77 (31.2%)
Easy data access	30 (23.8%)	24 (30.4%)	4 (9.5%)	58 (23.5%)
Overview	12 (9.5%)	20 (25.3%)	8 (19.1%)	40 (16.2%)
Improved stakeholder communication and relationship	13 (10.3%)	0	1 (2.4%)	14 (5.7%)
Accuracy of data	0	5 (6.3%)	6 (14.3%)	11 (4.5%)
Improved report creation	6 (4.8%)	0	3 (7.1%)	9 (3.6%)
User friendly	5 (4.0%)	0	2 (4.8%)	7 (2.8%)
Not applicable	2 (1.6%)	4 (5.1%)	1 (2.4%)	7 (2.8%)
Trends analysis views	6 (4.8%)	0	0	6 (2.4%)
Improved decision making and visibility	2 (1.6%)	2 (2.5%)	0	4 (1.6%)
Update frequency	0	4 (5.1%)	0	4 (1.6%)
Demand planning	3 (2.4%)	0	0	3 (1.2%)
Everything	2 (1.6%)	0	1 (2.4%)	3 (1.2%)
Supplier performance management	2 (1.6%)	0	0	2 (0.8%)
Management tool	1 (0.8%)	0	0	1 (0.4%)
Expiry date management	0	1 (1.3%)	0	1 (0.4%)

Nota Bene: respondents could have one or more favored functionality. PPE: Personal Protective Equipment.

**Table 6 healthcare-11-01838-t006:** Recommendations for each dashboard.

Recommendations for Each Dashboard	Integrated Medicine (n = 116)	COVID-19 (n = 74)	PPE (n = 39)	Total (n = 229)
None	38 (32.8%)	37 (50.0%)	16 (41.0%)	91 (39.7%)
Additional and standardised reports	10 (8.6%)	10 (13.5%)	11 (28.2%)	31 (13.5%)
Improve user functionality	21 (18.1%)	4 (5.4%)	1 (2.6%)	26 (11.4%)
More data points, e.g., batch/expiry	11 (9.5%)	9 (12.2%)	1 (2.6%)	21 (9.2%)
Improve data accuracy	15 (13.0%)	4 (5.4%)	0	19 (8.3%)
Standardisation of dashboards	9 (7.8%)	0	4 (10.3%)	13 (5.7%)
Improve adoption	3 (2.6%)	4 (5.4%)	2 (5.1%)	9 (4.0%)
Connectivity syncing or Rx API	4 (3.5%)	2 (2.7%)	1 (2.6%)	7 (3.1%)
Improving updating	2 (1.7%)	2 (2.7%)	2 (5.1%)	6 (2.6%)
User preferences	2 (1.7%)	0	0	2 (0.9%)
Decrease reporting frequency	0	1 (1.4%)	0	1 (0.4%)
Increase reporting frequency	0	0	1 (2.6%)	1 (0.4%)
Not applicable	0	1 (1.4%)	0	1 (0.4%)
Review key performance indicators	1 (0.9%)	0	0	1 (0.4%)

Nota Bene: respondents could have one or more favored functionality; Rx API: RxSolution application programming interface; PPE: Personal Protective Equipment.

**Table 7 healthcare-11-01838-t007:** Ways in which the NSC data improved medicines availability and monitoring.

Ways NSC Improved Medicines Availability and Monitoring	Medicines Availability (n = 142)	Monitoring Medicines Availability (n = 252)
Improved medicines availability and monitoring	75 (52.8%)	106 (42.1%)
Ease of access to data	14 (9.9%)	88 (34.9%)
More redistribution and visibility	17 (12.0%)	26 (10.3%)
Not my function or not sure	12 (8.6%)	14 (5.6%)
Improved data quality and quantity	4 (2.8%)	14 (5.6%)
Limited due to other issues	8 (5.6%)	4 (1.6%)
Proactive reducing effort	12 (8.5%)	0

**Table 8 healthcare-11-01838-t008:** NSC challenges at present.

NSC Challenges	Responses (n = 200)
Internet connectivity	53 (26.5%)
Data quality	40 (20.0%)
Frequency of data input	31 (15.5%)
Navigating the system	23 (11.5%)
Speed of the system	19 (9.5%)
Limited support when required	13 (6.5%)
None	12 (6.0%)
Feedback timeline	6 (3.0%)
User adoption and limited licences	2 (1.0%)

## Data Availability

Additional data are available from the corresponding author on reasonable request.

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
