# Peer review of "Perceptions of and Practical Experience with the National Surveillance Centre in Managing Medicines Availability Amongst Users within Public Healthcare Facilities in South Africa: Findings and Implications"

_healthcare, 2023, doi:10.3390/healthcare11131838_

Round 1

Reviewer 1 Report

Authors used a descriptive study aiming to “address knowledge gap and determine the role of the NSC (National Surveillance Center)” (L128). It is unclear whether this study has generalizability, but it may be important to confirm usefulness of newly developed system. However, this article has not been fully answered some of questions due to the insufficient description.

First, authors explain the purpose of this study as mentioned above, but this study focus on perception of users, but not their knowledge and their gap, and could not address knowledge gap. If authors aim to “address knowledge gap”, authors should use adequate study design.

Second, authors explain they aimed to determine the role of NSC, but government usually has plan including its role, before they get its budget, and the explanation that NSC was built without considering its role is questionable. Authors should investigate details of governmental plans including the role of NSC in budget proposal to explain the role of NSC.

Finally, there is no limitation in discussion section, but such questionnaire surveys usually have a lot of limitation. For example, perception of users may be biased toward positive results to avoid disappointment from investigators. Authors should add limitation of this study in discussion section.

Minor comments

L105. “SARS-CoV-2 virus” may be “SARS-CoV-2”.

L110. “coronary vascular disease” may be “cardiovascular disease” or “coronary heart disease”.

Table 2. “AMD” should be spelled out.

L105. “SARS-CoV-2 virus” may be “SARS-CoV-2”.

L110. “coronary vascular disease” may be “cardiovascular disease” or “coronary heart disease”.

Author Response

Reviewer 1

Open Review

Quality of English Language

( ) I am not qualified to assess the quality of English in this paper
( ) English very difficult to understand/incomprehensible
(x) Extensive editing of English language required
( ) Moderate editing of English language
( ) Minor editing of English language required
( ) English language fine. No issues detected

Author comments: Thank you for this. The paper has now been updated with the help of one of the co-authors who is a native English speaker with over 500 publications in peer-reviewed Journals to his name. We trust this is now acceptable.

Yes

Can be improved

Must be improved

Not applicable

Does the introduction provide sufficient background and include all relevant references?

( )

(x)

( )

( )

Are all the cited references relevant to the research?

( )

(x)

( )

( )

Is the research design appropriate?

( )

( )

(x)

( )

Are the methods adequately described?

( )

( )

(x)

( )

Are the results clearly presented?

( )

( )

(x)

( )

Are the conclusions supported by the results?

( )

( )

(x)

( )

Comments and Suggestions for Authors

  1. Authors used a descriptive study aiming to “address knowledge gap and determine the role of the NSC (National Surveillance Center)” (L128). It is unclear whether this study has generalizability, but it may be important to confirm usefulness of newly developed system. However, this article has not been fully answered some of questions due to the insufficient description.

Author comments: Thank you – we have now updated the paper and hope this is now acceptable.

  1. First, authors explain the purpose of this study as mentioned above, but this study focus on perception of users, but not their knowledge and their gap, and could not address knowledge gap. If authors aim to “address knowledge gap”, authors should use adequate study design.

Author comments: Thank you for this valuable comment. Indeed, we did not measure knowledge and have now rephrased the purpose of the study. As this was a descriptive study with a relatively small target population and subsequent sample size, it was not designed to be a KAP study. Furthermore, because perceptions are mostly explored in qualitative studies (which we did not do), we also replaced ‘perceptions’ with ‘perspectives’ in the title of the study and in the purpose, to cover the behavioural constructs that inform experience. We hope this is now clearer and acceptable.

  1. Second, authors explain they aimed to determine the role of NSC, but government usually has plan including its role, before they get its budget, and the explanation that NSC was built without considering its role is questionable. Authors should investigate details of governmental plans including the role of NSC in budget proposal to explain the role of NSC.

Author comments: Thank you for your comment. For your information, the NSC was developed and maintained in partnership with technical supporting partners which were donor funded with the future intentions of handing over to the department of health. Consequently, no budgeting has been required from the department of health for now. This will change. We have now included the fact that the NSC was initially donor funded in the updated paper and trust this is now OK.

  1. Finally, there is no limitation in discussion section, but such questionnaire surveys usually have a lot of limitation. For example, perception of users may be biased toward positive results to avoid disappointment from investigators. Authors should add limitation of this study in discussion section.

Author comments: Thank you for this comment. We have now added a paragraph at the end of the discussion recognizing the limitations of the study. We trust this is now OK.

Minor comments

  1. L105. “SARS-CoV-2 virus” may be “SARS-CoV-2”.

Author comments: I will change this.

  1. L110. “coronary vascular disease” may be “cardiovascular disease” or “coronary heart disease”.

Author comments: I will change this.

Table 2. “AMD” should be spelled out.

Author comments: Thank you, AMD has now been written out as ‘Affordable Medicines Directorate’. In addition, footnotes have been added to all tables and figures where applicable to explain abbreviations used. We trust this is now OK.

Comments on the Quality of English Language

L105. “SARS-CoV-2 virus” may be “SARS-CoV-2”.

L110. “coronary vascular disease” may be “cardiovascular disease” or “coronary heart disease”.

Author comments: Thank you for this. These changes have been made. In addition, the paper has now been updated with the help of one of the co-authors who is a native English speaker with over 500 publications in peer-reviewed Journals to his name. We trust this is now acceptable.

Reviewer 2 Report

Falco MF et al. has done a descriptive analysis to investigate HCP’s attitudes towards the national surveillance Center. This seems a key topic in South Africa due to the shortage of medicines and supplies. The methodologies and results are laid out clearly. A few aspects to be improved on are below:

1.     The supplementary table numbering needs to be fixed. There was no table S2, then suddenly table S9 appeared.

2.     This is a small sample survey study, please include the limitations for the study.

3.     The manuscript separates responders into three groups, is there any difference in the altitude to NSC between the three groups?

English edits:

Line 192: “enrolment” -> “enrollment”

Please edit the English on spelling and grammar. 

Author Response

Reviewer 2

Open Review

Quality of English Language

( ) I am not qualified to assess the quality of English in this paper
( ) English very difficult to understand/incomprehensible
( ) Extensive editing of English language required
( ) Moderate editing of English language
(x) Minor editing of English language required
( ) English language fine. No issues detected

Author comments: Thank you for this. The paper has now been updated with the help of one of the co-authors who is a native English speaker with over 500 publications in peer-reviewed Journals to his name. We trust this is now acceptable.

Yes

Can be improved

Must be improved

Not applicable

Does the introduction provide sufficient background and include all relevant references?

(x)

( )

( )

( )

Are all the cited references relevant to the research?

(x)

( )

( )

( )

Is the research design appropriate?

(x)

( )

( )

( )

Are the methods adequately described?

(x)

( )

( )

( )

Are the results clearly presented?

( )

(x)

( )

( )

Are the conclusions supported by the results?

( )

(x)

( )

( )

Comments and Suggestions for Authors

Falco MF et al. has done a descriptive analysis to investigate HCP’s attitudes towards the national surveillance Center. This seems a key topic in South Africa due to the shortage of medicines and supplies. The methodologies and results are laid out clearly.

Author comments: Thank you for these kind words – appreciated!

A few aspects to be improved on are below:

Author comments: Thank you for these – we hope we have adequately addressed these.

  1. The supplementary table numbering needs to be fixed. There was no table S2, then suddenly table S9 appeared.

Author comments. Thank you for this comment. All supplementary tables have now been included and referred to in the text as such, and we trust this is now OK

  1. This is a small sample survey study, please include the limitations for the study

Author comments: Thank you for this comment. A paragraph recognizing the limitations has been added to the discussion, and we hope this is now acceptable

.

  1. The manuscript separates responders into three groups, is there any difference in the altitude to NSC between the three groups?

Author comments: Thank you for this valuable comment. As this was a descriptive study with a relatively small target population and subsequent sample size across the groups (see point 2 above), it was not designed to be a KAP study and to compare the groups of respondents – rather to consolidate the different perspectives to provide guidance on how the system can be improved to address concerns with medicine shortages Furthermore, because perceptions are mostly explored in qualitative studies (which we did not do), we also replaced ‘perceptions’ with ‘perspectives’ in the title of the study and in the purpose, to cover the behavioural constructs that inform experience. We hope this is now clearer and acceptable.

  1. English edits:

Line 192: “enrolment” -> “enrollment”

Author comments: Now amended thank you

  1. Comments on the Quality of English Language. Please edit the English on spelling and grammar. 

Author comments: Thank you for this. The paper has now been updated with the help of one of the co-authors who is a native English speaker with over 500 publications in peer-reviewed Journals to his name. We trust this is now acceptable.

Reviewer 3 Report

1. Highlight the novelty of the paper

2. Give a diagram of the methodology, in the methodology section

3. To provide more details in section 2.3 by describing the analysis 

4. Check line 35, and make necessary corrections with respect to brackets

5. Reffered table S1 in line 169, is not available

Need careful check for minor mistakes.

Author Response

Reviewer 3

Open Review

Quality of English Language

( ) I am not qualified to assess the quality of English in this paper
( ) English very difficult to understand/incomprehensible
( ) Extensive editing of English language required
( ) Moderate editing of English language
(x) Minor editing of English language required
( ) English language fine. No issues detected

Author comments: Thank you for this. The paper has now been updated with the help of one of the co-authors who is a native English speaker with over 500 publications in peer-reviewed Journals to his name. We trust this is now acceptable.

Yes

Can be improved

Must be improved

Not applicable

Does the introduction provide sufficient background and include all relevant references?

(x)

( )

( )

( )

Are all the cited references relevant to the research?

(x)

( )

( )

( )

Is the research design appropriate?

( )

(x)

( )

( )

Are the methods adequately described?

( )

(x)

( )

( )

Are the results clearly presented?

(x)

( )

( )

( )

Are the conclusions supported by the results?

(x)

( )

( )

( )

Comments and Suggestions for Authors

  1. Highlight the novelty of the paper

Author comments: Thank you for this comment. We have now upgraded the comments at the end of the Introduction to state why we believe this study is both valuable and novel as South Africa seeks to improve knowledge regarding medicine availability in the public sector. We trust this is now acceptable

  1. Give a diagram of the methodology, in the methodology section

Author comments: Thank you – now inserted (new Figure 1).

  1. To provide more details in section 2.3 by describing the analysis 

Author comments: Thank you for this comment. Kindly note that this was a descriptive study, not including any inferential statistics. We have now revised the description of the data analysis and trust it is now acceptable.

  1. Check line 35, and make necessary corrections with respect to brackets

Author comments: Thank you – now updated. We have kept the brackets as they are with the other bracket design used for references. We trust this is now OK.

  1. Reffered table S1 in line 169, is not available

Author comments: Thank you for this comment. All supplementary tables (S1 to S5) have now been included at the end of the manuscript and are referred to in the text as such. Table S1 is in fact the Questionnaire – however as this is available online as a questionnaire we only included the link. We trust this is acceptable.

  1. Comments on the Quality of English Language - Need careful check for minor mistakes.

Thank you for this. The paper has now been updated with the help of one of the co-authors who is a native English speaker with over 500 publications in peer-reviewed Journals to his name. We trust this is now acceptable.

Round 2

Reviewer 1 Report

Authors revised the manuscript, but I still have minor comments.

Minor comments

L278-L279, L330-L331, L346-L348: It is difficult to understand what the abbreviations mean.

L285, L326, L346, L366, L550, L561: “NB” should be spelled out.

L278-L279, L330-L331, L346-L348: It is difficult to understand what the abbreviations mean.

L285, L326, L346, L366, L550, L561: “NB” should be spelled out.

Author Response

Reviewer 1

Open Review

Quality of English Language

( ) I am not qualified to assess the quality of English in this paper
( ) English very difficult to understand/incomprehensible
( ) Extensive editing of English language required
( ) Moderate editing of English language required
(x) Minor editing of English language required
( ) English language fine. No issues detected

Author comments: Thank you – appreciated!

Yes

Can be improved

Must be improved

Not applicable

Does the introduction provide sufficient background and include all relevant references?

(x)

( )

( )

( )

Are all the cited references relevant to the research?

(x)

( )

( )

( )

Is the research design appropriate?

(x)

( )

( )

( )

Are the methods adequately described?

(x)

( )

( )

( )

Are the results clearly presented?

(x)

( )

( )

( )

Are the conclusions supported by the results?

(x)

( )

( )

( )

Comments and Suggestions for Authors

Authors revised the manuscript, but I still have minor comments.

Thank you for your help. Hopefully, we have now addressed your additional minor comments.

Minor comments

L278-L279, L330-L331, L346-L348: It is difficult to understand what the abbreviations mean.

L285, L326, L346, L366, L550, L561: “NB” should be spelled out.

Author comments: Thank you – hopefully, this has now been addressed.

Comments on the Quality of English Language

L278-L279, L330-L331, L346-L348: It is difficult to understand what the abbreviations mean.

L285, L326, L346, L366, L550, L561: “NB” should be spelled out.

Author comments: Thank you – this has now been addressed.